# Application of transthoracic echocardiography to assess the dynamic evolution of early cardiac damage induced by abdominal aorta-inferior vena cava fistula in female rats with chronic renal failure

**Yiran Zhang**[ID][1], **Lizhou Wu**[1], **Liming Liang**[2], **Kuan Li**[3], **Xianglei Kong**[2]*, **Haiyan Wang**[ID][1]*

**1** Department of Medical Ultrasound, The First Affiliated Hospital of Shandong First Medical University & Shandong Provincial Qianfoshan Hospital, Jinan, Shandong, China, **2** Department of Nephrology, The First Affiliated Hospital of Shandong First Medical University & Shandong provincial Qianfoshan Hospital, Shandong institute of Nephrology, Jinan, Shandong, China, **3** Clinical Medical Institute, Xinjiang Medical University,Urumqi, China

* wanghaiyan96@126.com (HW); kxl1985@163.com (XK)

## Abstract

### Objective

Left ventricular dysfunction (LVD) frequently occurs in patients with chronic renal failure (CRF) who have an arteriovenous fistula (AVF). In a rat model of CRF with AVF, we assessed the utility of transthoracic echocardiography for the early detection of LVD and examined the associated pathological damage.

### Methods

Forty female rats that had successfully established a CRF model were divided into three groups: the CRF group (n = 13), the sham group (n = 13), and the AVF group (n = 14). The AVF was established (labeled as T0). Renal function and myocardial enzyme parameters were measured at T0, the 4th week (labeled as T1), and the 6th week (labeled as T2). The parameters of echocardiography were measured with an animal ultrasound device (Vevo 3100). The early diastolic peak flow velocity (E) was divided by the mitral valve's E wave deceleration time (DT), to calculate the E/DT ratio. TOMTEC image analysis software was utilized to analyze the LV's global longitudinal strain (GLS) and global circumferential strain (GCS). At each time point, three rats from each group were euthanized, and the left ventricular pathological tissues were collected for HE and Masson staining.

### Results

(1) At T1, the AVF group had no significant difference in GLS, although GCS and the E/DT ratio increased. At T2, the AVF showed lower GLS and GCS and a higher E/DT

**Data availability statement:** All relevant data are within the paper and its Supporting Information files.

**Funding:** This work was supported by : Natural Science Foundation of Shandong Province (ZR2023MH041).

**Competing interests:** The authors have declared that no competing interests exist.

ratio. (2) Significant alterations were observed in AVF group tissues stained with HE and Masson at T1 and T2.

## Conclusions

This study found that pathological damage to the left ventricular myocardium persisted after the rat model was established. Unlike traditional echocardiography measurements, GLS, GCS, and the E/DT ratio can detect dynamic changes in left ventricular function.

## Introduction

Hemodialysis (HD) increases the survival time of patients with end-stage renal disease (ESRD). In 2017, approximately three million patients received HD; by 2030, this amount is predicted to rise to 5.4 million [1]. By the end of 2020, China's HD patient population had surpassed 690,000. Although HD medication can help ESRD patients live longer lifetimes, their mortality risk remains much higher than that of the general population [2]. Cardiovascular disease causes approximately 40% of mortality in dialysis-dependent ESRD patients [2,3]. AVF is widely utilized to gain vascular access for HD. According to research, implementing of AVF can aggravate cardiovascular strain and potentially lead to myocardial infarction [4,5]. As a result, doctors face difficulty in detecting and intervening early in cases of AVF-induced heart injury. Globally, most animal models of CRF involve procedures like 5/6 nephrectomy or adenine supplementation [6]. However, the development of an autologous AVF model is still in its early stages, and research into the cardiac pathological damage produced by AVF-induced CRF and its causes is limited.

The purpose of this study is to reproduce the pathological features of early cardiac damage in rats with CRF by constructing an AVF. In addition, multiple echocardiographic techniques were utilized to dynamically analyze cardiac shape and function in the animal model as early as possible, indicating the evolution of early cardiac damage caused by AVF.

## Materials

### Animal research

All methods followed the rules stated in the European Parliament's Directive 2010/63/EU on the protection of animals used for scientific purposes, as well as the NIH Guide for the Care and Use of Laboratory Animals. The study was approved by the Bioethics Committee of the First Affiliated Hospital of the Shandong First Medical University [No. 2021 (S1066)] and "Provincial Natural Animal Ethics [2022] Animal Ethics Review No. (S758)". All procedures were performed with isoflurane anesthesia (5% induction and 1.5% maintenance, with a 1.5 L O2/min flow rate of isoflurane), and every attempt was made to reduce pain. Forty-five female *Sprague-Dawley* (SD) rats (purchased from Jinan Pengyue Experimental Animal Breeding Co., Ltd., 6–8 weeks old, 250–300g) were housed in a temperature-controlled room (21°C–24°C and

60% relative humidity, with a 12-hour light-dark cycle) with unrestricted access to water, and all rats were acclimated for 3 days to ensure the adaptation to the new environment. Following the American Veterinary Medical Association Guidelines for the Euthanasia of Animals (2020), all rats were sacrificed humanely via cervical dislocation after anesthesia with isoflurane.

### Establishment of the chronic renal failure model

The rats' renal function (blood urea nitrogen, BUN; serum creatinine, Scr) was measured. All rats (n = 45) were fed standard pellet feed containing 0.75% adenine (Beijing Keao Xielixi Feed Co., Ltd.) for 4 weeks. The adenine feed contained 0.63% phosphorus, 0.74% calcium, 0.53% potassium, and 0.22% sodium. To confirm the operational efficacy of the CRF model (n = 40), BUN and Scr levels were reviewed after four weeks of feeding, and body weight (BW) measurements were conducted on the rats. The rats were subsequently switched to a regular diet for continuing feeding [7–9].

### Grouping

A total of 40 rats (40/45, 88.89%) with CRF were randomly assigned to three groups: the CRF group (n = 13), the sham group (n = 13), and the AVF group (n = 14).

### Arteriovenous fistula creation

The rats were sedated with isoflurane (5% induction and 1.5% maintenance, with a 1.5 L O2/min flow rate of isoflurane) using a small animal anesthesia machine (RWD, China). Following anesthesia induction, the rats were allowed to breathe freely, while their vital signs were monitored. The animals were positioned supine on a heating pad. The abdominal area was cleaned, and the Abdominal Aorta (AA) and Inferior vena cava (IVC) were revealed. A 3 mm longitudinal incision was created on the AA and IVC. The wound was closed using a continuous, side-to-side suture method. To establish hemostasis, the artery clamps were released, and mild pressure was applied to the anastomosis site. The IVC was loaded with blood, that was changing from dark red to light red, and there were discernible vibrations. The jet-like flow of arterial blood into the IVC (labeled as T0) indicated that the AVF had been successfully created. After confirming that there was no considerable active bleeding, the abdominal cavity was closed layer by layer, and the skin was sutured (with a 3−0 nylon thread for continuous suturing)[10]. The rats in the sham group were sedated before having their AA and IVC examined. T1 and T2 time points were defined as four and six weeks after AVF formation, respectively [9].

### Monitoring parameters

Renal function and myocardial enzymes (creatine kinase, CK; creatine kinase MB Form, CK-MB; lactate dehydrogenase, LDH) were measured by collecting tail vein blood samples at T0, T1, and T2 for each of the three groups. After allowing 2 mL of blood per rat to spontaneously clot at room temperature for 10–20 min, the samples were centrifuged at 3000 rpm for 10 min. The supernatant was collected and analyzed with an automated biochemical analyzer (Hitachi 7170). Scr: Quantified using the creatinine enzymatic technique; BUN: Measured by urease-glutamate dehydrogenase coupled assay. CK: Measured using the enzyme-coupled UV rate assay; CK-MB: Determined through the immunoinhibition method; LDH: Assessed via the lactate-to-pyruvate substrate conversion. BW measurements were performed on the three experimental groups at T0, T1, and T2 time points.

### Ultrasound parameters

**Ultrasound parameters of AVF.**  Under anesthesia, the inner diameter of the AVF in rats was measured using a high-resolution small-animal ultrasound imaging system (Vevo 3100 FUJIFILM Visual Sonics, Toronto, Canada). The fistula was localized via color Doppler flow imaging (CDFI), and the peak systolic velocity (PSV) and flow volume were

quantified using pulsed-wave Doppler (PWD) at the anastomotic site. Measurements were performed 2 mm proximal to the anastomosis, with a sample volume of 1.5 mm and a Doppler angle <60°. All hemodynamic parameters were averaged over three consecutive cardiac cycles.

**Echocardiographic parameters.** The rat's cardiac anatomy and function were assessed at T0, T1, and T2 using ultrasound device while sedated. The ultrasound probe was placed on the left side of the sternum to get the left ventricular long-axis view and measure the left atrial diameter (LAD). In the left ventricular long-axis view, M-mode ultrasound was used to capture the motion curve and takes measurements of the left ventricular. The analysis included the following parameters: heart rate (HR), left ventricular end-diastolic diameter (LVEDD), left ventricular end-systolic diameter (LVESD), left ventricular end-diastolic volume (LVEDV), left ventricular end-systolic volume (LVESV), interventricular septal thickness (IVS), left ventricular posterior wall thickness (LVPW), left ventricular mass (LVM), left ventricular ejection fraction (LVEF), left ventricular short-axis fractional shortening (LVFS), left ventricular cardiac output (LVCO), and right ventricular cardiac output (RVCO).

$$LVEF(\%) = \frac{(LVEDV - LVESV) \times 100}{LVEDV}$$

$$LVM(mg) = 1.055 \times \left[ (IVS + LVEDD + LVPW)^3 - LVIDd^3 \right]$$

$$LVFS(\%) = \frac{(LVEDD - LVESD) \times 100}{LVEDD}$$

In the apical four-chamber view, the sample volume was placed at the mitral valve orifice. Using pulsed-wave Doppler, measure the E wave, A wave, and the DT, then calculate the E/A ratio and E/DT ratio. Under the same modality, measure the left ventricular isovolumic relaxation time (IRT, ms), isovolumic contraction time (ICT, ms), and ejection time (ET, ms) to compute the Tei index. Each parameter was recorded over for three consecutive heartbeats and then averaged.

$$Tei \text{ Index} = \frac{(ICT + IRT)}{ET}$$

The TOMTEC image analysis software (version 2.31; TOMTEC Imaging Systems GmbH, Germany) was utilized for offline analysis of images to quantify the left ventricular GLS and GCS in each rat group. However, the evaluation was confined to the left cardiac system because the right atrium and ventricle were hidden due to the rat's narrow chest structure and intercostal spaces. Measurements of the inner diameter and forward flow velocity of the fistula were taken at T0, T1, and T2 to determine the fistula blood flow. To reduce the potential variability in cardiac function assessments, two experienced echocardiographers, who were unaware of the study details, independently conducted echocardiographic examinations, and the average of their two measurements was calculated.

## Histopathological examination

Three rats from each of the CRF, sham, and AVF groups were humanely sacrificed right after the cardiac ultrasound parameters were obtained at T0 and T1, respectively. The rest of the rats were humanely sacrificed at T2. Anesthesia was administered using isoflurane (5% induction and 1.5% for maintenance, with a 1.5 L $O_2$/min flow rate of isoflurane) through a small animal anesthesia machine (RWD, China) and humane sacrifice was performed through cervical dislocation. The left side of each rat's thoracic cavity was opened to expose the heart, and a small piece of left ventricular myocardial

tissue was removed. This tissue was then fixed in a 4% paraformaldehyde solution, dehydrated in ethanol, embedded in paraffin, and continuously selection. The sections were deparaffinized, cleared in xylene, and stained with HE and Masson to observe histopathological changes in the left ventricular myocardium.

## Statistical methods

The experimental data were analyzed using the SPSS software, version 26.0. The measurement data were represented as mean±SD. For comparisons among multiple groups, a one-way analysis of variance was employed, and independent sample t-tests were used for pairwise comparisons. A significance level of $P<0.05$ was considered statistically significant.

## Results

### Establishment the of chronic renal failure and AVF models in rats

This study involved 45 rats. After being fed with 0.75% adenine for 4 weeks, the CRF model was successfully developed in 40 rats, showing significantly elevated BUN and Scr ($P=0.046$; $P=0.004$). The success rate was 88.89%. AVF surgery was successfully performed on 14 of the CRF rats to create the AVF model. Unfortunately, two rats died after surgery, leading to a survival rate of 85.7%. During the subsequent pathological autopsy, rats in the AVF group exhibited complications such as heart failure, ascites, abdominal infection, intestinal torsion, and abdominal hematoma.

### Changes in myocardial enzymes and renal function parameters in the three groups of rats

**Changes in myocardial enzyme levels.** At T0, there were no significant statistical differences in CK, CK-MB, and LDH levels among the three groups ($\eta^2=0.1$, $P=0.980$; $P=0.476$; $P=0.607$). However, at T1, the levels of CK, CK-MB, and LDH in the AVF group were higher than those in the CRF group ($P=0.005$; $P=0.028$; $P<0.001$) and the sham group ($P=0.002$; $P=0.029$; $P<0.001$). By T2, the levels of CK, CK-MB, and LDH in the AVF group were significantly elevated compared to those in the CRF group ($P=0.001$; $P=0.028$; $P<0.001$) and the sham group ($P=0.001$; $P=0.001$; $P<0.001$). The complete myocardial enzyme parameters of rats can be found in S1 File.

**Changes in renal function indicators.** There were no significant statistical differences in BUN level among the CRF group, the sham group and the AVF group at T0 ($P=0.433$), T1 ($P=0.856$) and T2 ($P=0.467$). Similarly, there were no significant statistical differences in Scr levels among the CKD group, the sham group and the AVF group at T0 ($P=0.399$), T1 ($P=0.320$) and T2 ($P=0.560$). The complete renal function parameters of rats can be found in S1 File.

### Changes in BW in the three groups of rats

As shown in Table 1, there were no significant statistical variations in BW among the CRF group, the sham group and the AVF group at T0, T1 and T2. (Table 1) The raw BW data of rats can be found in S1 File.

### Changes in ultrasound parameters of fistula at different time points

In the AVF group rats, the changes in the inner diameter and blood flow of the fistula were not statistically significant at T0, T1, and T2 ($P=0.712$; $P=0.392$) (Table 2). The raw parameters of internal diameter and flow rate of the fistula in the AVF group are detailed in S2 File.

### Changes in echocardiographic parameters among the three groups at different time points

At T0: No significant differences were found among the three groups.

At T1: Compared to the CRF and sham groups, the AVF group exhibited statistically significant differences in LAD, LVEDD, LVEDV and LVM ($P=0.037$; $P=0.047$; $P=0.001$; $P=0.001$). The IVS and LVPW thickness increased ($P=0.019$; $P=0.004$). At T1, the E/DT ratio increased ($P=0.001$) (Fig 1a), but there were no significant statistical differences in the

**Table 1. Changes in renal function parameters, myocardial enzyme levels and BW in the rat CRF model.**

| Renal function parameters | the CRF group 95% confidence interval [CI]: | the sham group 95% confidence interval [CI]: | the AVF group 95% confidence interval [CI]: |
|---|---|---|---|
| BUN (mg/dl) | &$P$=0.52, η²=0.047 | &$P$=0.454, η²=0.057 | &$P$=0.852, η²=0.013 |
| T0 *$P$=0.433, η²=0.054 | 13.16±5.59 (n=13) [10.05, 16.41] | 14.76±4.29 (n=13) [12.30, 17.19] | 15.86±4.60 (n=14) [13.21, 18.55] |
| T1 *$P$=0.856, η²=0.011 | 15.44±4.45 (n=10) [12.96, 18.07] | 16.41±4.79 (n=10) [13.65, 19.33] | 16.39±4.69 (n=9) [13.52, 19.27] |
| T2 *$P$=0.4967, η²=0.073 | 14.94±4.06 (n=7) [12.31, 17.83] | 17.39±4.70 (n=7) [14.25, 20.91] | 17.08±3.80 (n=6) [13.89, 19.81] |
| Scr (μmol/L) | &$P$=0.872, η²=0.01 | &$P$=0.890, η²=0.009 | &$P$=0.785, η²=0.019 |
| T0 *$P$=0.399, η²=0.059 | 73.25±21.50 (n=13) [60.64, 86.27] | 77.80±19.64(n=13) [66.46, 89.16] | 86.09±25.02(n=14) [71.45, 100.09] |
| T1 *$P$=0.32, η²=0.076 | 74.83±18.91 (n=10) [64.27, 87.07] | 80.58±18.20 (n=10) [70.18, 91.68] | 88.39±23.46(n=9) [73.11, 102.62] |
| T2 *$P$=0.56, η²=0.056 | 78.10±19.33 (n=7) [65.00, 92.04] | 83.46±38.24 (n=7) [58.83, 112.23] | 94.40±26.40(n=6) [75.14, 114.83] |
| Myocardial enzyme | the CRF group 95% confidence interval [CI]: | the sham group 95% confidence interval [CI]: | the AVF group 95% confidence interval [CI]: |
| CK (U/L) | &$P$=0.007, η²=0.307 | &$P$=0.001, η²=0.585 | &$P$<0.001, η²=0.714 |
| T0 *$P$=0.98, η²=0.01 | 176.40±23.15 (n=13) [163.60, 188.84] | 174.03±24.09 (n=13) [159.22, 190.07] | 166.05±28.14(n=14) [155.83, 196.30] |
| T1 *$P$<0.001, η²=0.552 | 203.94±29.80 (n=10) [184.71, 219.66] | 190.64±32.17 (n=10) [171.07, 211.05] | 317.60±83.15*#▲(n=9) [267.84, 370.49] |
| T2 *$P$=0.001, η²=0.494 | 267.06±102.57▲♦(n=7) [212.87, 348.38] | 272.31±51.62▲♦(n=7) [240.39, 307.51] | 437.63±90.12*#▲♦(n=6) [365.76, 495.58] |
| CK-MB (U/L) | &$P$=0.03, η²=0.353 | &$P$<0.001, η²=0.573 | &$P$<0.001, η²=0.666 |
| T0 *$P$=0.476, η²=0.048 | 186.85±46.70 (n=13) [159.78, 215.24] | 199.35±35.56 (n=13) [178.37, 219.52] | 177.19±43.52(n=14) [152.23, 204.41] |
| T1 *$P$=0.044, η²=0.194 | 244.12±60.15▲(n=10) [209.12, 277.38] | 244.42±51.67 (n=10) [215.90, 276.41] | 323.04±112.98*#▲(n=9) [263.48, 398.39] |
| T2 *$P$=0.005, η²=0.412 | 301.34±88.28▲(n=7) [248.62, 376.06] | 375.11±101.12▲♦(n=7) [311.39, 450.75] | 504.65±127.15*#▲♦(n=6) [409.94, 600.26] |
| LDH (U/L) | &$P$=0.004, η²=0.337 | &$P$<0.001, η²=0.449 | &$P$=<0.001, η²=0.834 |
| T0 *$P$=0.607, η²=0.033 | 185.48±23.37 (n=13) [169.67, 202.60] | 174.03±24.09 (n=13) [159.22, 190.08] | 166.05±28.14(n=14) [149.30, 181.63] |
| T1 *$P$<0.001, η²=0.771 | 203.94±29.80 (n=10) [184.71, 219.66] | 204.55±37.92 (n=10) [183.90, 228.55] | 386.60±70.49*#▲(n=9) [341.51, 429.99] |
| T2 *$P$<0.001, η²=0.770 | 242.06±41.54▲♦(n=7) [212.64, 272.25] | 277.31±77.29▲♦(n=7) [236.21, 340.99] | 581.92±128.13*#▲♦(n=6) [502.36, 690.65] |
| BW (g) | &$P$=0.809, η²=0.016 | &$P$=0.437, η²=0.059 | &$P$=0.974, η²=0.02 |
| T0 *$P$=0.809, η²=0.011 | 278.62±8.08(n=13) [275.00, 282.33] | 276.69±7.33 (n=13) [274.00, 279.50] | 276.86±11.31(n=14) [270.89, 283.10] |
| T1 *$P$=0.641, η²=0.034 | 280.51±7.08(n=10) [276.14, 284.67] | 277.70±6.70(n=10) [273.33, 282.01] | 277.67±8.78(n=9) [272.03, 283.11] |
| T2 *$P$=0.658, η²=0.048 | 279.29±6.29(n=7) [274.63, 283.87] | 280.71±8.28(n=7) [274.68, 286.20] | 276.50±10.11(n=6) [269.97, 285.67] |

(BUN, blood urea nitrogen; Scr, serum creatinine; CK, creatine kinase; CK-MB, creatine kinase-MB; LDH, lactate dehydrogenase.

*$P$<0.05 for intergroup comparisons at each time point;

&$P$<0.05 for intragroup comparisons across T0, T1, and T2.

*$P$<0.05 compared to the CRF group;

#$P$<0.05 compared to the sham group;

▲$P$<0.05 compared to T0;

♦$P$<0.05 compared to T1.)

**Table 2. Changes in the fistula diameter and flow rate in the AVF group after the establishment of the AVF model.**

| indexes | T0 | T1 | T2 |
|---|---|---|---|
| Fistula Diameter(mm) | 1.27±0.20 (n=14) [1.14, 1.40] | 1.33±0.25 (n=9) [1.14, 1.53] | 1.34±0.19 (n=6) [1.26, 1.43] |
| Flow Rate(uL) | 106.94±34.07 (n=14) [84.05, 129.82] | 127.36±49.63 (n=9) [89.21, 165.51] | 131.51±34.42 (n=6) [95.39, 167.64] |

(There were no statistically significant differences in the fistula diameter and flow rate at T0, T1, and T2 [$P=0.712$, $\eta^2=0.029$; $P=0.392$, $\eta^2=0.078$])

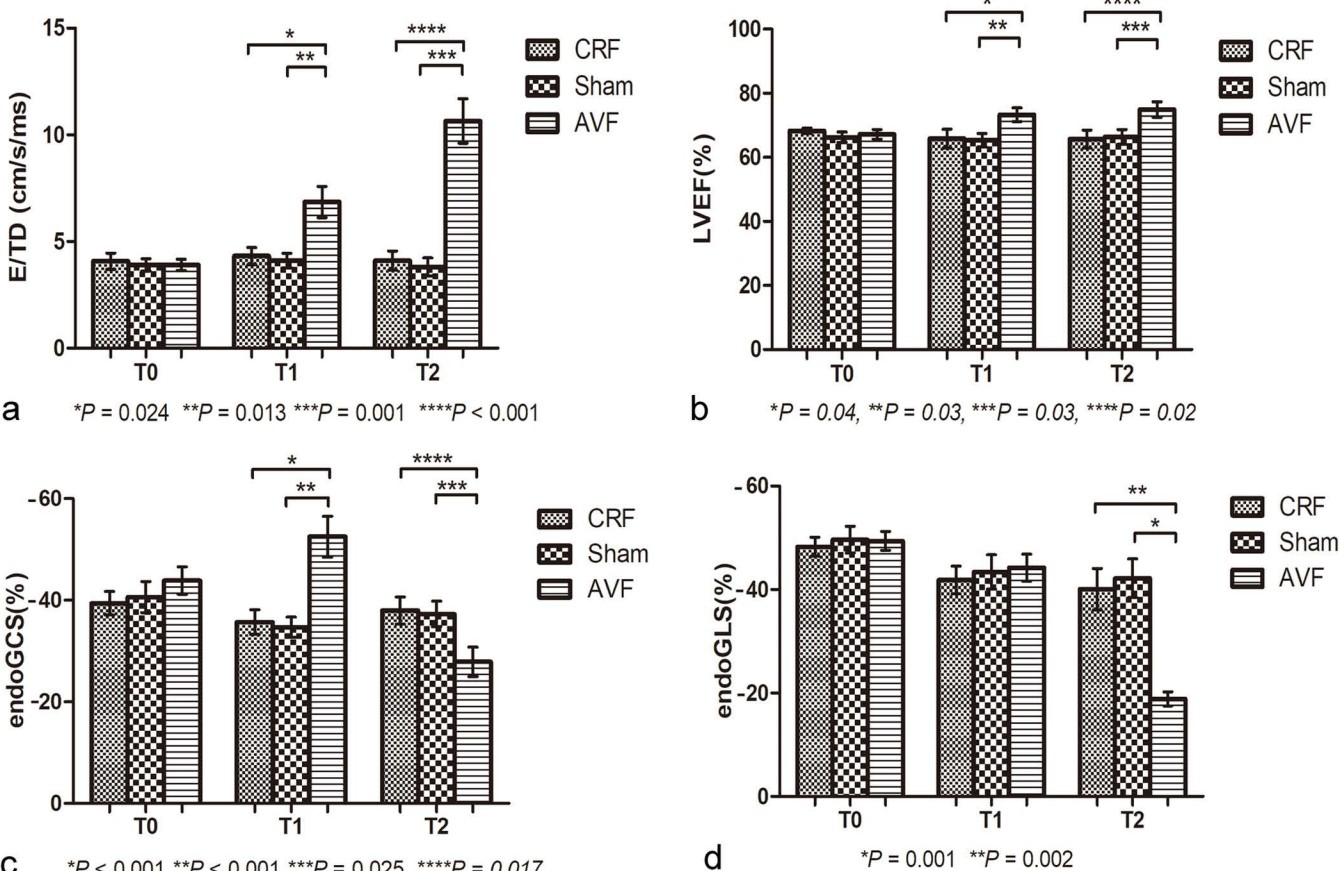

**Fig 1. a:** At 4 weeks postoperatively, the E/DT values in the AVF group (n = 9, 6.58±2.71 cm/s/ms) were significantly higher than those in the CRF group (n = 100, 4.32±1.6 cm/s/ms, P = 0.024) and the sham group (n = 100, 4.111±1.08 cm/s/ms, P = 0.013). At 6 weeks postoperatively, the E/DT values in the AVF group (n = 6, 10.65±2.9 cm/s/ms) were significantly higher than those in the CRF group (n = 7, 4.10±1.19 cm/s/ms, P =0.001) and the sham group (n = 7, 3.811±1.14 cm/s/ms, P < 0.001). **b:** At 4 weeks postoperatively, the LVEF values in the AVF group (n = 9, 73.32%±6.79%) were significantly higher than those in the CRF group n = 10, 65.84%±9.73%, P = 0.04) and the sham group (n= 10, 65.39%±6.86%, P = 0.05). At 6 weeks postoperatively, the LVEF values in the AVF group ( n= 6, 74.96%±6.37%) were significantly higher than those in the CRF group (n = 7, 65.71%±8.92%, P = 0.03) and the sham group (n = 7, 66.40%±6.63%, P = 0.02). **c:** At 4 weeks postoperatively, GCS absolute value in the AVF group (n = 9, 57.50%±9.40%) was significantly higher than that in the CRF group (n = 10, 35.70%±8.00%, P < 0.001) and the sham group (n = 10, 34.68%±6.54%, P < 0.001), while at 6 weeks postoperatively, GCS absolute value in the AVF group (n = 6, 27.89%±7.67%) was significantly lower than that in the CRF group (n = 7, 37.95%±7.62%, P = 0.017) and the sham group (n = 7, 37.29%±7.14%, P = 0.025). **d:** At 6 weeks postoperatively, GLS absolute value in the AVF group (n = 6, 18.89%±3.60%) was significantly lower than that in the CRF group (n = 7, 40.08%±11.29%, P = 0.002) and the sham group (n = 7, 42.19%±10.57%, P = 0.001).

Tei index for the left and right ventricles ($P=0.775$; $P=0.310$), and there were no significant statistical differences in the E/A ratios of the mitral and tricuspid valves ($P=0.189$; $P=0.297$). The AVF group showed higher left ventricular systolic function, there were statistical differences in LVEF and LVFS ($P=0.048$; $P=0.001$) (Fig 1b). The LVCO and RVCO were significantly elevated ($P=0.036$; $P<0.001$). The GCS increased ($P<0.001$) (Fig 1c), whereas there were no significant changes in the GLS ($P=0.843$) (Fig 1d). There were no significant statistical differences in the HR ($P=0.585$).

At T2: Compared to the CRF and sham groups, the AVF group exhibited further significant increases in LAD, LVEDD, LVEDV and LVM ($P=0.001$; $P=0.001$; $P<0.001$; $P<0.001$). IVS and LVPW thickened further ($P<0.001$; $P=0.003$). The E/DT ratio increased significantly ($P<0.001$) (Fig 1a). There were no significant statistical differences in the left and right ventricular Tei index ($P=0.155$; $P=0.221$), and there were no significant statistical differences in the E/A ratios of the mitral and tricuspid valves ($P=0.170$; $P=0.842$). LVEF and LVFS increased ($P=0.039$; $P=0.013$) (Fig 1b). LVCO and RVCO significantly increased ($P<0.001$; $P=0.012$). GCS decreased further ($P=0.031$) (Fig 1c); GLS decreased significantly ($P<0.001$) (Fig 1d).. An increase in HR was observed ($P<0.001$). At T1 and T2, compared with the sham group, there were no statistically significant differences in the above data.

**Changes in echocardiographic parameters within the three groups at different time points**

At the time points T0, T1, and T2, there were no statistically significant differences in the echocardiographic parameters between the CRF and sham groups ($P>0.05$).

In the AVF group at T1, compared to T0, there were increases in LAD, LVEDD, LVEDV and LVM ($P=0.029$; $P=0.019$; $P=0.006$; $P=0.001$), thickening of IVS and LVPW ($P<0.001$; $P=0.002$), and increases in LVEF and LVFS ($P=0.029$; $P=0.021$). Additionally, LVCO and RVCO increased ($P=0.032$; $P<0.001$). Although the GCS was elevated ($P=0.002$), the GLS exhibited no statistically significant alterations ($P=0.081$), representative images are shown in Fig 2 and Fig 3. The E/DT ratio was also increased ($P=0.026$). However there were no statistically significant differences in the Tei index for the left and right ventricles ($P=0.869$; $P=0.556$).Similarly, there were no significant differences in the E/A ratios of the mitral and tricuspid valves ($P=0.935$; $P=0.845$).

In the AVF group at T2 compared to T0, the parameters showed the following changes: LAD, LVEDD, LVEDV and LVM further increased ($P<0.001$), IVS and LVPW significantly thickened ($P<0.001$), LVEF and LVFS significantly increased ($P=0.014$; $P=0.003$), and LVCO and RVCO significantly increased ($P<0.001$; $P=0.001$). GCS and GLS decreased ($P=0.001$; $P<0.001$), and the E/DT ratio increased ($P=0.001$). There were no statistically significant differences in the Tei index for the left and right ventricular ($P=0.352$; $P=0.086$) or the E/A ratio of the mitral and tricuspid valves ($P=0.403$; $P=0.362$).

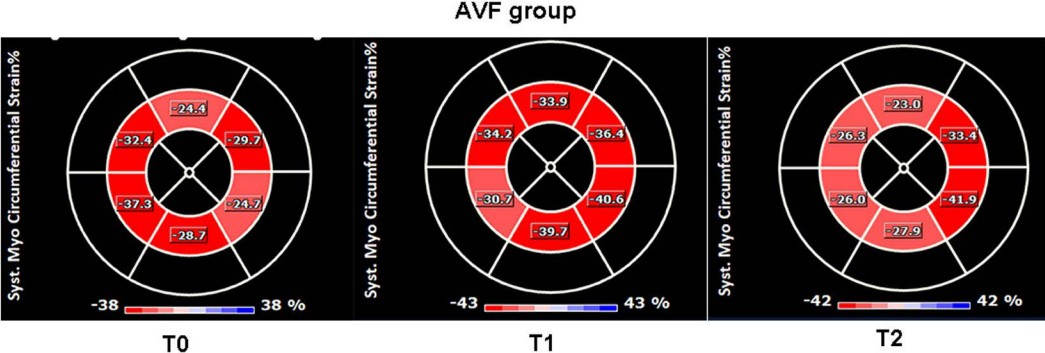

**Fig 2. Changes in the left ventricular GCS value in the AVF group at T0, T1, and T2 in the short axis view of the left ventricle.**

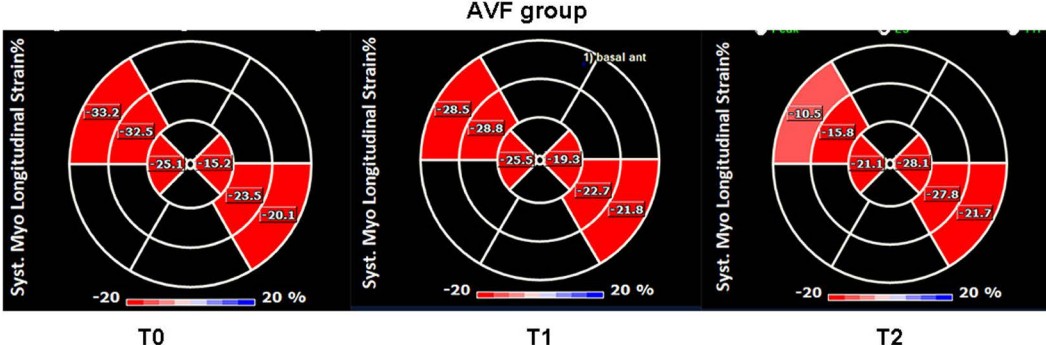

**Fig 3. Changes in the left ventricular GLS value in the AVF group at T0, T1, and T2 in the long axis view of the left ventricle.**

Changes in the various parameters in the AVF group at T2 compared to T1 were as follows: the LAD, LVEDD, LVEDV and LVM increased ($P=0.037$; $P=0.043$; $P=0.045$; $P=0.013$), IVS thickened ($P=0.024$), LVPW exhibited no significant change ($P=0.914$), LVEF and LVFS showed no statistically significant differences ($P=0.586$; $P=0.305$); LVCO and RVCO increased ($P=0.007$; $P=0.041$), GCS and GLS significantly decreased ($P<0.001$), and the E/DT ratio increased ($P=0.024$). There were no significant statistical differences in the left and right ventricular Tei index ($P=0.442$; $P=0.233$) or the E/A ratios of the mitral and tricuspid valves ($P=0.372$; $P=0.469$) (Table 3). The raw echocardiographic data for all rats can be found in S3 File.

## Myocardial pathological changes

Histopathological analysis was performed on the left ventricular myocardial tissues of the three groups of rats at T0, T1, and T2, respectively.

At T0, the myocardial tissues of the three groups of rats demonstrated enlarged myocardial cells that were deeply stained. There was a slight presence of infiltrating inflammatory cells visible under HE staining. However, when examined with Masson staining, there was no notable increase in fibrous tissue.

By T1, the myocardial tissues obtained from the CRF and sham group rats displayed several pathological features, including enlarged myocardial cells and nuclei, deep staining, myocardial cell interstitial edema, and infiltration of inflammatory cells on HE staining. Masson staining revealed an increase in fibrous tissue formation beneath the endocardium. In addition to these pathological changes observed in the CRF/sham group, the AVF group showed enlarged myocardial cells and nuclear rupture (Fig 4). There was a significant increase in fibrous tissue within the extracellular matrix and around small blood vessels, as demonstrated by HE and Masson staining.

By T2, the myocardial tissues of the AVF group rats showed increased formation of fibrous tissue beneath the endocardium. Additionally, Masson staining revealed rupture and dissolution of myocardial fibers (Fig 5).

## Discussion

Cardiovascular disease is a major complication among patients with ESRD who are undergoing HD. However, there is a lack of extensive research on the mechanisms behind cardiac remodeling following the creation of AVF. This study revealed the following findings: (1) At T1, initiation of left ventricular remodeling was observed in the AVF group, with increased GCS, and a minor increase in the E/DT ratio. (2) At T2, LV remodeling in the AVF group became more evident, with further reductions in GLS and GCS, along with a continued increase in the E/DT ratio. (3) Long-term AVF patency led to myocardial cell swelling and even necrosis, an increase in fibrous tissue as demonstrated by Masson staining, as well as myocardial fiber rupture and dissolution.

**Table 3. Changes in echocardiographic parameters in the three groups of rats.**

| Echocardiographic Parameters | the CRF group [95% confidence intervals] | the sham group [95% confidence intervals] | the AVF group [95% confidence intervals] |
|---|---|---|---|
| GCS (%) | $^{\&}P=0.54$, $\eta^2=0.045$ | $^{\&}P=0.25$, $\eta^2=0.097$ | $^{\&}P<0.001$, $\eta^2=0.649$ |
| T0 *$P=0.49$, $\eta^2=0.046$ | $-39.40\pm7.64$ (n=13) [-34.94, -44.10] | $-40.60\pm10.11$ (n=13) [-35.02, -46.82] | $-43.87\pm8.99$ (n=14) [-39.37, -49.23] |
| T1 *$P<0.001$, $\eta^2=0.648$ | $-35.70\pm8.00$ (n=10) [- 31.31, - 40.22] | $-34.68\pm6.54$ (n=10) [-31.04, -39.03] | $-57.50\pm9.40$*#▲ (n=9) [-52.04, -63.10] |
| T2 *$P=0.031$, $\eta^2=0.293$ | $-37.95\pm7.62$ (n=7) [-32.90,- 42.60] | $-37.29\pm7.14$ (n=7) [-32.60, -42.20] | $-27.89\pm7.67$*#▲♦ (n=6) [-22.26, -33.66] |
| GLS (%) | $^{\&}P=0.105$, $\eta^2=0.154$ | $^{\&}P=0.219$, $\eta^2=0.106$ | $^{\&}P<0.001$, $\eta^2=0.800$ |
| T0 *$P=0.888$, $\eta^2=0.08$ | $-48.26\pm6.12$ (n=13) [-44.52,- 51.73] | $-49.64\pm8.59$ (n=13) [-43.99,- 54.55] | $-49.39\pm6.13$ (n=14) [-45.89,- 52.88] |
| T1 *$P=0.843$, $\eta^2=0.12$ | $-41.84\pm8.86$ (n=10) [-36.71,- 47.73] | $-43.38\pm11.03$ (n=10) [-36.41,- 49.78] | $-44.21\pm8.24$ (n=9) [-39.15,- 49.71] |
| T2 *$P<0.001$, $\eta^2=0.581$ | $-40.08\pm11.29$ (n=7) [-31.77,- 48.31] | $-42.19\pm10.57$ (n=7) [-35.30,- 49.82] | $-18.89\pm3.60$*#▲♦ (n=6) [-16.05,- 21.50] |
| E/DT (cm/s/ms) | $^{\&}P=0.894$, $\eta^2=0.008$ | $^{\&}P=0.840$, $\eta^2=0.013$ | $^{\&}P<0.001$, $\eta^2=0.639$ |
| T0 *$P=0.908$, $\eta^2=0.05$ | $4.08\pm1.37$ (n=13) [3.44, 4.71] | $3.911\pm1.05$ (n=13) [3.27, 4.55] | $3.91\pm0.97$ (n=14) [3.29, 4.52] |
| T1 *$P=0.001$, $\eta^2=0.373$ | $4.3\pm1.26$ (n=10) [3.21, 5.44] | $4.1\pm1.08$ (n=10) [2.99, 5.22] | $6.85\pm2.71$*#▲ (n=9) [5.79, 7.92] |
| T2 *$P<0.001$, $\eta^2=0.747$ | $4.11\pm1.20$ (n=7) [2.51, 5.70] | $3.81\pm1.14$ (n=7) [2.21, 5.40] | $10.65\pm2.95$*#▲♦ (n=6) [9.16, 12.14] |
| RV Tei index | $^{\&}P=0.299$, $\eta^2=0.086$ | $^{\&}P=0.264$ $\eta^2=0.094$ | $^{\&}P=0.219$, $\eta^2=0.114$ |
| T0 *$P=0.607$, $\eta^2=0.033$ | $0.20\pm0.06$ (n=13) [0.18, 0.31] | $0.15\pm0.03$ (n=13) [0.08, 0.21] | $0.19\pm0.06$ (n=14) [0.12, 0.25] |
| T1 *$P=0.31$, $\eta^2=0.078$ | $0.16\pm0.05$ (n=10) [0.13, 0.21] | $0.17\pm0.04$ (n=10) [0.15, 0.18] | $0.23\pm0.05$ (n=9) [0.14, 0.34] |
| T2 *$P=0.221$, $\eta^2=0.140$ | $0.25\pm0.07$ (n=7) [0.18, 0.34] | $0.20\pm0.06$ (n=7) [0.17, 0.23] | $0.32\pm0.08$ (n=6) [0.17, 0.45] |
| RVCO (ml) | $^{\&}P=0.210$, $\eta^2=0.109$ | $^{\&}P=0.465$, $\eta^2=0.055$ | $^{\&}P<0.001$, $\eta^2=0.751$ |
| T0 *$P=0.179$, $\eta^2=0.108$ | $76.93\pm14.98$ (n=13) [68.39, 85.77] | $65.88\pm11.84$ (n=13) [55.17, 75.23] | $69.14\pm8.68$ (n=14) [63.70, 73.53] |
| T1 *$P<0.001$, $\eta^2=0.735$ | $64.8\pm17.86$ (n=10) [54.48, 75.13] | $59.36\pm12.78$ (n=10) [52.39, 67.71] | $153.66\pm42.14$*#▲ (n=9) [129.00, 179.40] |
| T2 *$P<0.001$, $\eta^2=0.829$ | $74.45\pm15.84$ (n=7) [56.28, 81.89] | $67.83\pm15.69$ (n=7) [56.28, 81.89] | $232.86\pm60.97$*#▲♦ (n=6) [187.34, 276.97] |
| LAD (mm) | $^{\&}P=0.456$, $\eta^2=0.057$ | $^{\&}P=0.824$, $\eta^2=0.014$ | $^{\&}P=0.001$, $\eta^2=0.432$ |
| T0 *$P=0.758$, $\eta^2=0.018$ | $4.10\pm0.30$ (n=13) [3.91, 4.28] | $4.20\pm0.41$ (n=13) [3.97, 4.43] | $4.23\pm0.52$ (n=14) [3.94, 4.54] |
| T1 *$P=0.037$, $\eta^2=0.203$ | $4.29\pm0.57$ (n=10) [3.95, 4.64] | $4.23\pm0.30$ (n=10) [4.06, 4.42] | $4.84\pm0.75$*#▲ (n=9) [4.43, 5.33] |
| T2 *$P=0.001$, $\eta^2=0.494$ | $4.39\pm0.62$ (n=7) [3.91, 4.75] | $4.32\pm0.56$ (n=7) [3.95, 4.74] | $5.49\pm0.48$*#▲♦ (n=6) [5.13, 5.86] |
| LV Tei index | $^{\&}P=0.155$, $\eta^2=0.129$ | $^{\&}P=0.254$, $\eta^2=0.097$ | $^{\&}P=0.622$, $\eta^2=0.037$ |
| T0 *$P=0.501$, $\eta^2=0.045$ | $0.20\pm0.05$ (n=13) [0.13, 0.28] | $0.16\pm0.03$ (n=13) [0.11, 0.19] | $0.21\pm0.05$ (n=14) [0.14, 0.29] |
| T1 *$P=0.775$, $\eta^2=0.017$ | $0.18\pm0.04$ (n=10) [0.14, 0.22] | $0.20\pm0.05$ (n=10) [0.16, 0.23] | $0.22\pm0.06$ (n=9) [0.13, 0.29] |
| T2 *$P=0.155$, $\eta^2=0.17$ | $0.27\pm0..07$ (n=7) [0.21, 0.33] | $0.20\pm0.06$ (n=7) [0.18, 0.23] | $0.28\pm0.06$ (n=6) [0.21, 0.34] |
| LVFS (%) | $^{\&}P=0.33$, $\eta^2=0.079$ | $^{\&}P=0.674$, $\eta^2=0.029$ | $^{\&}P=0.007$, $\eta^2=0.328$ |

*(Continued)*

| Echocardiographic Parameters | the CRF group [95% confidence intervals] | the sham group [95% confidence intervals] | the AVF group [95% confidence intervals] |
|---|---|---|---|
| T0 *P=0.839, η²=0.012 | 43.23±5.70 (n=13) [39.37, 47.04] | 44.91±9.38 (n=13) [38.61, 51.20] | 44.66±6.05 (n=14) [40.59, 48.73] |
| T1 *P=0.001, η²=0.362 | 46.45±4.03 (n=10) [43.74, 49.16] | 42.13±4.73 (n=10) [38.95, 45.31] | 52.65±8.45*#▲ (n=9) [46.60, 58.69] |
| T2 *P=0.013, η²=0.353 | 46.94±8.38 (n=7) [39.94, 53.94] | 43.68±6.91 (n=7) [37.90, 49.46] | 56.47±7.81*#▲ (n=6) [52.81, 71.28] |
| LVEF (%) | &P=0.708, η²=0.025 | &P=0.996, η²=0.02 | &P=0.024, η²=0.259 |
| T0 *P=0.594, η²=0.034 | 68.33±5.73 (n=13) [64.47, 72.18] | 66.33±5.37 (n=13) [62.73, 69.94] | 67.20±5.06 (n=14) [63.81, 70.61] |
| T1 *P=0.044, η²=0.183 | 65.84±9.73 (n=10) [59.31, 72.38] | 65.39±6.86 (n=10) [60.78, 70.00] | 73.32±6.79*#▲ (n=9) [68.46, 70.61] |
| T2 *P=0.039, η²=0.278 | 65.71±8.92 (n=7) [58.95, 72.48] | 66.40±6.63 (n=7) [60.86, 71.94] | 74.96±6.37*#▲ (n=6) [69.07, 80.85] |
| LVPW (mm) | &P=0.143, η²=0.134 | &P=0.22, η²=0.106 | &P<0.001, η²=0.603 |
| T0 *P=0.554, η²=0.039 | 0.91±0.29 (n=13) [0.71, 1.10] | 0.99±0.07 (n=13) [0.94, 1.03] | 0.93±0.07 (n=14) [0.88, 0.98] |
| T1 *P=0.04, η²=0.313 | 1.07±0.14 (n=10) [0.98, 1.16] | 1.06±0.10 (n=10) [0.99, 1.12] | 1.28±0.22*#▲ (n=9) [1.12, 1.44] |
| T2 *P=0.003, η²=0.448 | 1.08±0.18 (n=7) [0.93, 1.23] | 1.04±0.13 (n=7) [0.94, 1.06] | 1.33±0.14*#▲ (n=6) [1.20, 1.46] |
| LVEDD (mm) | &P=0.59, η²=0.038 | &P=0.485, η²=0.052 | &P=0.001, η²=0.447 |
| T0 *P=0.375, η²=0.063 | 7.87±0.57 (n=13) [7.48, 8.26] | 7.94±0.77 (n=13) [7.43,8.46] | 7.53±0.78 (n=14) [7.01, 8.06] |
| T1 *P=0.047, η²=0.191 | 7.63±0.84 (n=10) [7.07, 8.20] | 7.56±0.76 (n=10) [7.05, 8.07] | 8.48±1.05*#▲ (n=9) [7.72, 9.23] |
| T2 *P=0.001, η²=0.496 | 8.00±0.84 (n=7) [7.28, 8.68] | 7.57±0.94 (n=7) [6.79,8.36] | 9.37±0.62*#▲◆ (n=6) [8.81, 9.95] |
| LVESD (mm) | &P=0.262, η²=0.094 | &P=0.271, η²=0.092 | &P=0.511, η²=0.050 |
| T0 *P=0489, η²=0.038 | 4.47±0.48 (n=13) [4.18, 4.76] | 4.42±0.47 (n=13) [4.16, 4.70] | 4.65.±0.58(n=14) [4.31, 4.98] |
| T1 *P=0.587, η²=0.040 | 4.32±0.57 (n=10) [3.91, 4.72] | 4.50±0.49(n=10) [4.14, 4.85] | 4.55±0.49(n=9) [4.18, 4.93] |
| T2 *P=0.210, η²=0.168 | 4.75±0.52 (n=7) [4.27, 5.22] | 4.77±0.36(n=7) [4.44, 5.10] | 4.34±0.51(n=6) [3.80, 4.87] |
| LVEDV (ml) | &P=0.926, η²=0.02 | &P=0.263, η²=0.094 | &P<0.001, η²=0.500 |
| T0 *P=0.865, η²=0.01 | 250.43±27.8 1(n=13) [231.75, 269.12] | 257.44±32.24 (n=13) [235.79, 279.10] | 252.78±32.78 (n=14) [230.76, 274.80] |
| T1 *P=0.001, η²=0.394 | 244.61±47.46 (n=10) [212.73, 276.50] | 235.48±25.74 (n=10) [218.19, 252.77] | 330.35±80.04*#▲ (n=9) [273.10, 387.61] |
| T2 *P<0.001, η²=0.729 | 249.76±33.14 (n=7) [222.05, 277.46] | 252.76±39.20 (n=7) [219.99, 285.53] | 391.28±53.42*#▲◆ (n=6) [341.89, 440.69] |
| LVESV (ml) | &P=0.69, η²=0.027 | &P=0.001, η²=0.384 | &P<0.001, η²=0.447 |
| T0 *P=0.299, η²=0.063 | 83.17±11.55(n=13) [76.19,90.15] | 89.77±9.46(n=13) [84.06,95.49] | 84.44±12.68(n=14) [77.12, 91.76] |
| T1 *P=0.603, η²=0.038 | 77.95±19.23(n=10) [64.20,91.71] | 71.25±9.94(n=10) ▲ [64.15,78.36] | 71.76±18.40(n=9) [57.61, 85.90] |
| T2 *P=0.005, η²=0.466 | 82.18±12.30(n=7) [70.80,93.55] | 81.08±13.89(n=7) [68.23,93.93] | 110.35±19.54(n=6) *#▲◆[89.84, 130.86] |
| IVS (mm) | &P=0.704, η²=0.026 | &P=0.396, η²=0.066 | &P<0.001, η²=0.651 |
| T0 *P=0.581, η²=0.036 | 1.06±0.16 (n=13) [0.95, 1.16] | 1.04±0.10 (n=13) [0.98, 1.11] | 1.00±0.10 (n=14) [0.94, 1.07] |

*(Continued)*

**Table 3.** (Continued)

| Echocardiographic Parameters | the CRF group [95% confidence intervals] | the sham group [95% confidence intervals] | the AVF group [95% confidence intervals] |
|---|---|---|---|
| T1 *$P$=0.019, η²=0.24 | 1.09±0.13 (n=10) [1.01, 1.18] | 1.11±0.15 (n=10) [1.01, 1.22] | 1.29±0.19*#▲ (n=9) [1.15, 1.43] |
| T2 *$P$ <0.001, η²=0.652 | 1.11±0.14 (n=7) [0.99, 1.22] | 1.10±0.13 (n=7) [0.99, 1.20] | 1.46±0.12*#▲♦ (n=6) [1.35, 1.57] |
| LVCO (ml) | &$P$=0.908, η²=0.007 | &$P$=0.890, η²=0.009 | &$P$=0.001, η²=0.505 |
| T0 *$P$=0.630, η²=0.030 | 62.46±7.85 (n=13) [57.18, 67.73] | 61.53±12.12 (n=13) [53.39, 69.67] | 58.42±10.28 (n=14) [51.52, 65.33] |
| T1 *$P$=0.036, η²=0.205 | 62.10±12.77 (n=10) [53.52, 70.67] | 61.28±9.45 (n=10) [54.94, 67.64] | 90.48±46.67*#▲ (n=9) [57.09, 123.87] |
| T2 *$P$<0.001, η²=0.776 | 64.28±12.62 (n=7) [53.72, 74.83] | 63.61±11.68 (n=7) [53.85, 73.37] | 137.44±30.34*#▲♦ (n=6) [109.38, 165.50] |
| mitral valve E/A | &$P$=0.278, η²=0.091 | &$P$=0.164, η²=0.125 | &$P$=0.621, η²=0.037 |
| T0 *$P$=0.858, η²=0.001 | 1.18±0.40 (n=13) [0.95, 1.42] | 1.26±0.39 (n=13) [1.03, 1.48] | 1.19±0.29 (n=14) [1.02, 1.35] |
| T1 *$P$=0.189, η²=0.109 | 0.94±0.26 (n=10) [0.79, 1.11] | 1.02±0.29 (n=10) [0.85, 1.19] | 1.11±0.33 (n=9) [0.99, 1.41] |
| T2 *$P$=0.170, η²=0.162 | 1.07±0.32 (n=7) [0.84, 1.28] | 1.15±0.21 (n=7) [0.81, 1.21] | 1.33±0.42 (n=6) [1.05, 1.68] |
| tricuspid valve E/A | &$P$=0.938, η²=0.005 | &$P$=0.917, η²=0.006 | &$P$=0.639, η²=0.035 |
| T0 *$P$=0.369, η²=0.064 | 0.91±0.36 (n=13) [0.67, 1.15] | 0.92±0.22 (n=13) [0.77, 1.07] | 1.08±0.33 (n=14) [0.86, 1.30] |
| T1 *$P$=0.297, η²=0.080 | 0.93±0.17 (n=10) [0.81, 1.05] | 0.85±0.27 (n=10) [0.70, 1.06] | 1.02±0.30 (n=9) [0.83, 1.28] |
| T2 *$P$=0.842, η²=0.017 | 0.88±0.25 (n=7) [0.73, 1.04] | 0.90±0.21 (n=7) [0.74, 1.10] | 0.95±0.17 (n=6) [0.79, 1.11] |
| HR (bmp) | &$P$=0.973, η²=0.02 | &$P$=0.399, η²=0.066 | &$P$=0.003, η²=0.36 |
| T0 *$P$=0.534, η²=0.032 | 347.92±15.96(n=13) [338.28, 357.57] | 347.15±9.81 (n=13) [341.22, 353.08] | 343.14±9.40(n=14) [337.72, 348.57] |
| T1 *$P$=0.585, η²=0.040 | 346.80±8.55 (n=10) [340.69, 352.92] | 348.70±7.87(n=10) [343.07, 354.33] | 337.67±42.10 (n=9) [305.31, 370.03] |
| T2 *$P$<0.001, η²=0.732 | 347.86±7.29 (n=7) [341.12, 354.60] | 343.01±6.48 (n=7) [337.01, 348.99] | 385.83±19.87 (n=6) *#▲♦[364.98, 406.69] |
| LVM (mg) | &$P$=0.239, η²=0.101 | &$P$=0.137, η²=0.137 | &$P$<0.001, η²=0.624 |
| T0 *$P$=0.681, η²=0.021 | 982.64±64.06(n=13) [943.93, 1021.35] | 1001.86±67.80(n=13) [960.88, 1042.83] | 977.21±91.03(n=14) [924.65, 1029.77] |
| T1 *$P$=0.001, η²=0.401 | 977.81±94.42 (n=10) [910.27, 1045.36] | 982.05±38.20(n=10) [954.72, 1009.37] | 1179.27±165.64(n=9) *#▲[1051.95, 1306.60] |
| T2 *$P$<0.001 η²=0.816 | 1035.30±50.91 (n=7) [988.21, 1082.38] | 1040.15±57.79(n=7) [986.70, 1093.61] | 1350.88±106.86(n=6) *#▲♦[1238.73, 1463.02] |

(GCS, global circumferential strain; GLS, global longitudinal strain; E/DT, the mitral valve early diastolic peak flow velocity (E)/ E wave deceleration time (DT); RV, right ventricular; LV, left ventricular; RVCO, right ventricular cardiac output; LAD, left atrial diameter; LVFS, left ventricular short-axis fractional shortening; LVEF, left ventricular ejection fraction; LVPW, left ventricular posterior wall thickness; LVEDD, left ventricular end-diastolic diameter; LVESD, left ventricular end-systolic diameter; LVEDV, left ventricular end-diastolic volume; LVESV, left ventricular end-systolic volume; IVS, interventricular septal thickness; LVCO, left ventricular cardiac output; HR, heart rate; LVM, left ventricular mass.

*$P$<0.05 for intergroup comparisons at each time point;

&$P$<0.05 for intragroup comparisons across T0, T1, and T2.

*$P$<0.05 compared to the CRF group;

#$P$<0.05 compared to the sham group;

▲$P$<0.05 compared to T0;

♦$P$<0.05 compared to T1.)

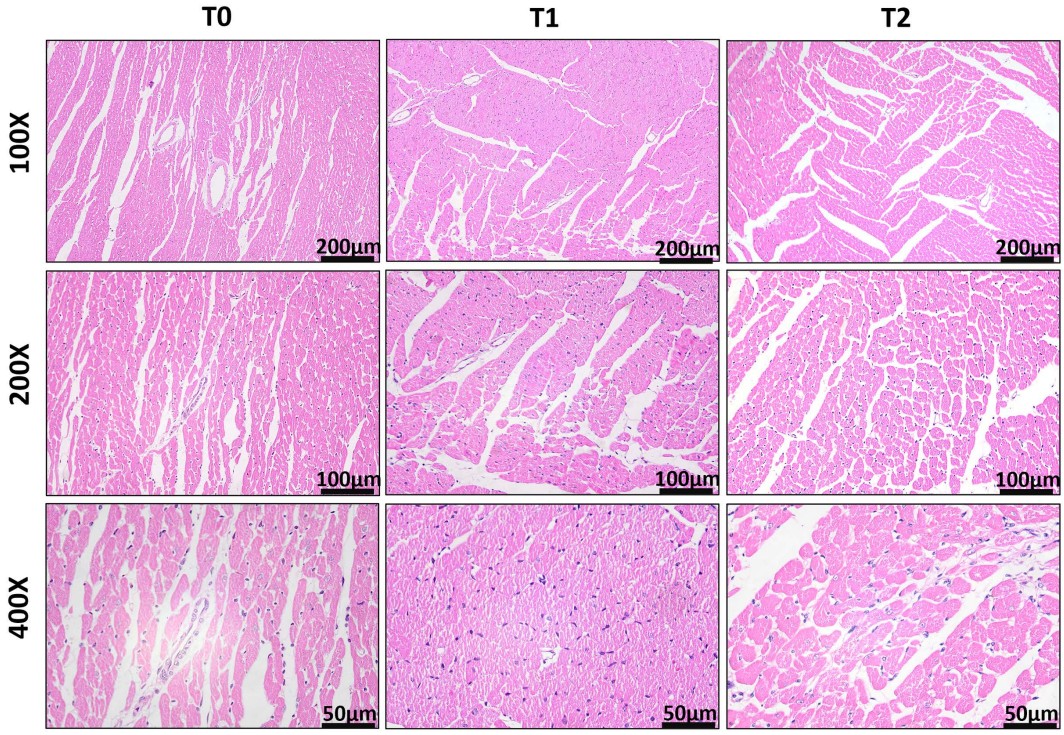

**Fig 4. HE staining of the left ventricular myocardial tissues in the AVF group at T0, T1, and T2.**

After the creation of AVF, there is a notable rise in the filling volume of the left and right sides of the heart, as well as CO [11]. These hemodynamic changes trigger the activation of the neuroendocrine system, including the renin-angiotensin-aldosterone system (RAAS). In the early stages following AVF creation, angiotensin II (Ang II) can provoke the enlargement of heart muscle cells, which leads to myocardial fibrosis and cell death. This process ultimately contributes to ongoing cardiac remodeling and a decline in heart function [12].In this study, researchers successfully. developed a rat model of CRF induced by AVF. Echocardiography revealed ongoing left ventricular remodeling, including left ventricular hypertrophy (LVH). Additionally, there was an increase in left ventricular myocardial contractility and LVEF. The creation of an AVF resulted in more significant histopathological changes in the LV myocardial tissue, such as increased interstitial and perivascular fibrosis within the LV myocardium, along with cardiomyocyte necrosis. Persistent oxidative stress, activation of the RAAS, ongoing inflammation, and myocardial ischemia may contribute to progressive myocardial fibrosis, leading to greater ventricular stiffness and diastolic filling restriction. Moreover, LVH extends the isovolumetric relaxation time, elevates myocardial oxygen consumption, decreases left ventricular compliance, raises filling pressure, and further worsens left ventricular diastolic dysfunction. In a study using the C57BL/6 mice AVF model, the cardiac changes observed, such as CO, LVEDD, LVEDV, and LVESV exhibited a trend similar to our study [13], whereas the LVEF showed a significant decline. This discrepancy might be attributed to the earlier onset of ventricular dilatation observed in the mice AVF model, which could to a deterioration in the LVEF. Furthermore, extending the follow-up duration in our study might also reveal a decline in LVEF. Clinical studies conducted by Momeni A et al. demonstrated a significant reduction in LVEF 12 months following the creation of an AVF [14]. Studies by Brower et al. [15] demonstrated that after creating an AVF between the AA and IVC, only 3% of rats experienced a significant reduction in EF and FS. However, approximately 50% of the rats showed an increase in lung mass within 8 weeks after the AVF was created. Chen et al. [16] found that this increase in lung mass was a sign of LVD leading to pulmonary

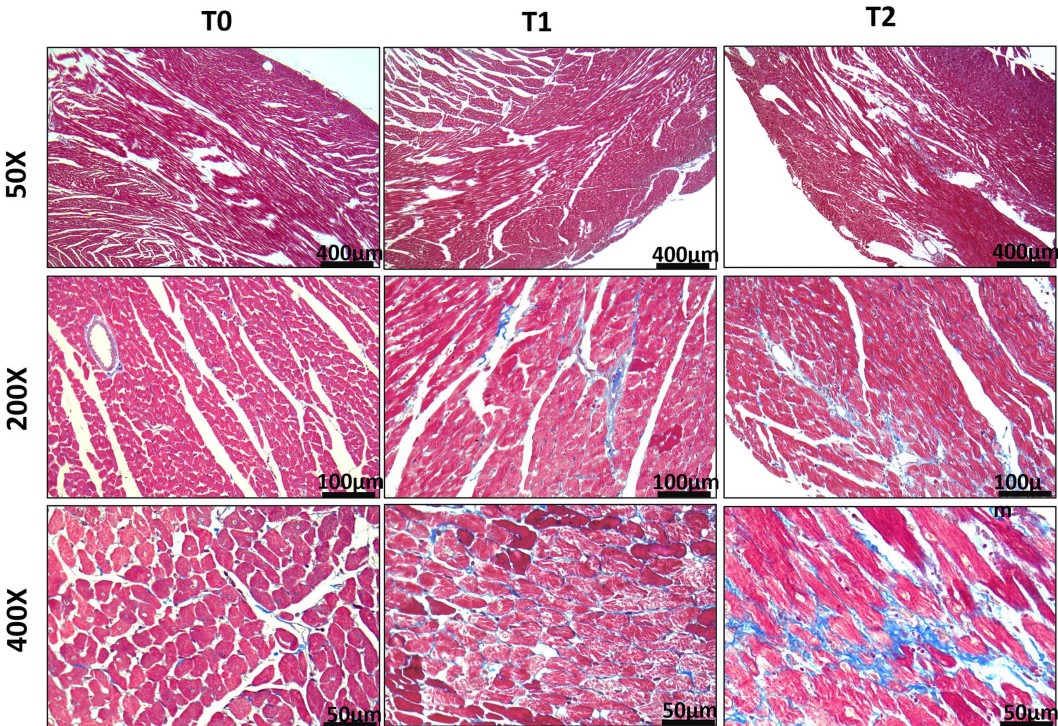

**Fig 5. Masson staining of the left ventricular myocardial tissues in the AVF group at T0, T1, and T2.**

congestion. This suggests that increased lung mass indirectly indicates LVD, whereas traditional echocardiographic parameters are not sensitive indicators of cardiac function in rats with an AVF. Thus, there is need for more sensitive, noninvasive echocardiographic parameters for evaluation.

Strain imaging (SI) employs two-dimensional speckle tracking to measure the longitudinal, radial, and circumferential motion of the myocardial fibers using two-dimensional grayscale images. This technique captures real-time movement and deformation of the myocardium to investigate global and regional systolic and diastolic functions [17]. SI is capable of detecting early LVD with greater sensitivity than traditional echocardiography [18]. Among the parameters derived from SI, longitudinal strain is the best indicator of myocardial shortening at the endocardial level and is considered the most sensitive marker of subendocardial myocardial fiber dysfunction [17]. As a result, longitudinal strain is viewed as the earliest sign of cardiac dysfunction. The mathematical model developed by. Stokke et al.[19] showed that myocardial fiber longitudinal shortening has a moderate role in maintaining EF within the normal range, whereas circumferential shortening plays a more significant role in modulating EF. This model further suggested that even when longitudinal and circumferential strains are reduced, the EF may remain within normal limits. In the current study, the AVF group also showed a decrease in GLS and GCS at T2, whereas LVEF remained within the normal range. Savage et al. [20] conducted studies using tonometry to collect peripheral arterial pressure waveforms from patients with CRF, those undergoing HD, and kidney transplant recipients. After the creation of AVF, the subendocardial viability ratio (SEVR) has been consistently observed to decrease in patients undergoing HD, highlighting a substantial effect of AVF formation on subendocardial perfusion [20]. Therefore, theoretically, after AVF creation, LV GLS would decrease. However, several clinical studies have indicated that LV GCS, which is a sensitive indicator of mid-layer circumferential myocardial contractile activity, changes before a significant reduction in GLS occurs, with this change appearing as a compensatory enhancement [21].This conclusion aligns with the findings by Pappritz et al. [22], who investigated type

I diabetes-associated myocardial disease in rats with streptozotocin (STZ) and assessed myocardial deformation using speckle tracking technology. They discovered that in the early stages of the type I diabetes animal model (6 weeks), GLS decreased while GCS increased. In our study, we noted an increase in GCS and no significant change in GLS in the AVF group at T1. However, histopathological examination revealed left ventricular myocardial cell hypertrophy, inflammation infiltration, and fibrosis in certain areas of the myocardial interstitium. This suggests that myocardial damage was present in the left ventricle at T1. At T2, GCS and GLS had decreased, indicating that with the prolonged establishment of the AVF, there was a deterioration in the contractile performance of the entire myocardial layer, aligning with more severe pathological changes.

The mitral inflow spectral Doppler curve is currently the most widely used and fundamental technique for assessing left ventricular diastolic function in conventional echocardiography [23]. The primary parameters include the E and A waves, the E/A ratio, DT, and the E/e ratio. As the mitral inflow Doppler is load-dependent, these parameters may display "pseudo-normalization [24]. In recent years, it has been discovered that the E/DT ratio may be more effective than other conventional parameters in predicting adverse clinical events in certain disease states. Nguyen et al. [25] investigated pulmonary congestion and remodeling in an HFpEF animal model and found that the E/DT ratio effectively reflects the pulmonary mass/body mass ratio of HFpEF mice. The correlation between the E/DT ratio and the pulmonary mass/body mass ratio was strong ($r = 0.76$, $P < 0.0001$). Therefore, the E/DT ratio may serve asa more convenient indicator for predicting secondary pulmonary remodeling in HFpEF. In a large-scale study involving 3,102 American Indians with hypertension and diabetes followed over an 8 years, Mishra et al. found that, after excluding other confounding factors, the DT/E ratio (HR, 1.09; 95% CI, 1.00–1.18; $P = 0.04$ for every 0.89 msec/[cm/s]) or the E/DT ratio could effectively predict fatal and nonfatal cardiovascular events, whereas DT or E alone had no predictive value [26]. Theoretically, in the initial stage of diastolic dysfunction, the left ventricular myocardium's active relaxation diminishes, resulting in a reduced E wave, an extended DT, and a lower E/DT ratio. As the condition advances, left ventricular stiffness increases, causing an enlarged E wave and a shortened DT. Thus, the higher the E/DT ratio, the more severe the left ventricular diastolic dysfunction. Diastolic dysfunction is quite prevalent among patients undergoing HD and serves as an independent predictor of mortality [27]. Our study findings showed that the E/DT ratio progressively increased over time in the AVF model of CRF rats, suggesting that the severity of left ventricular diastolic dysfunction intensified as the disease progressed.

It is well recognized that the process of cardiac remodeling may involve sex-specific differences, which could introduce potential confounding factors in the results [28]. In this study, female rats were utilized for the following reasons: the current research is built upon previous work by our team, which focused on sex-specific responses in developing an AVF model [10]. Female rats were used to ensure methodological continuity. However, we fully acknowledged the importance of sex-based differences in cardiac pathophysiology. Future studies will specifically include male rats to address this crucial aspect.

## Limitations

First, we attempted to induce CRF using a 0.75% adenine diet. However, this approach may have led to pathological manifestations similar to acute kidney injury. We plan to optimize the induction method in future studies. Second, we set our study's endpoint at 6 weeks after AVF establishment due to the significant increase in the mortality rate among rats. However, this limitation prevented us from obtaining further results on the changes in cardiac function and myocardial pathology resulting from long-term AVF maintenance. Third, because our study only included female rats, we overlooked the influence of sex. Due to current limitations in equipment and technical capabilities, we did not measure cardiomyocyte dimensions in this study, unlike in previous research [29]. We plan to address this methodological gap in future investigations by incorporating advanced cellular morphometric analyses. Lastly, the relatively small sample size may render the results of this study susceptible to inherent bias.

## Conclusions

In summary, this study discovered that CRF rats in the AVF model showed ongoing structural and functional cardiac damage. Compared to traditional ultrasound parameters, SI parameters obtained through two-dimensional speckle tracking echocardiography and the E/DT ratio may offer an earlier indication of the dynamic process of cardiac damage.

## Supporting information

**S1 File. Raw data of renal function parameters, myocardial enzyme parameters, and BW.**
(XLSX)

**S2 File. Raw data of the fistula diameter and flow rate.**
(XLSX)

**S3 file. Raw data of chocardiographic parameters.**
(XLSX)

## Acknowledgments

We sincerely acknowledge Dr. Wenhui Song for providing substantial preliminary clinical groundwork for this study and participating in early-stage data collection.

## Author contributions

**Data curation:** yiran Zhang, Liming Liang, Kuan Li.

**Formal analysis:** yiran Zhang, Lizhou Wu, Kuan Li.

**Writing – original draft:** yiran Zhang.

**Writing – review & editing:** Xianglei Kong, Haiyan Wang.

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
