## [Decision Letter · Decision Letter 0]

18 Feb 2025

PONE-D-24-47138Application of Transthoracic Echocardiography to Assess the Dynamic Evolution of Early Cardiac Damage Induced by Abdominal Aorta-Inferior Vena Cava Fistula in Rats with Chronic Renal FailurePLOS ONE

Dear Dr. Wang,

Thank you for submitting your manuscript to PLOS ONE. After careful consideration, we feel that it has merit but does not fully meet PLOS ONE’s publication criteria as it currently stands. Therefore, we invite you to submit a revised version of the manuscript that addresses the points raised during the review process.

**ACADEMIC EDITOR: ** All issues raised by editor and expert reviewers are required and should be addressed carefully.

We look forward to receiving your revised manuscript.

Kind regards,

Vincenzo Lionetti, M.D., PhD

Academic Editor

PLOS ONE

Journal Requirements:

2. To comply with PLOS ONE submissions requirements, in your Methods section, please provide additional information regarding the experiments involving animals and ensure you have included details on (1) methods of sacrifice, and (2) efforts to alleviate suffering.

3. Thank you for stating the following financial disclosure: [This work was supported by : Natural Science Foundation of Shandong Province (ZR2023MH041)]. 

Additional Editor Comments:

Major issues should be carefully addressed by the authors:

1) Title: add "...in female rats..." and discuss in the main text the choice of female rats.

2) "All surgery was performed under isoflurane...": please add concentration of the dose

3) Add information on assay used to measure each parameter.

4) Add reference supporting the CRF model and add more methodological details on the model.

5) Is echocardiography performed in awaken or anesthetized animals?

6) TOMTEC image analysis software: add details on the firm.

7) Table 1: Is the CRF model without significant changes in BUN and serum creatinine a clinically relevant model? Please discuss on it mentioning proper reference

8) Table 3: add data on heart rate and cardiac mass.

9) Echocardiographic parameters should be normalized on body weight

10) Figure legends: improve the text by adding sample size.

11) Since CRF leads to hypertrophic cardiomyocytes, the authors should measure size of cardiomyocytes. Finally, data should be discussed in the light of previous unmentioned study(10.1126/sciadv.adj4846).

Reviewers' comments:

Reviewer's Responses to Questions

**Comments to the Author**

1. Is the manuscript technically sound, and do the data support the conclusions?

Reviewer #1: Yes

Reviewer #2: Yes

2. Has the statistical analysis been performed appropriately and rigorously? 

Reviewer #1: Yes

Reviewer #2: Yes

3. Have the authors made all data underlying the findings in their manuscript fully available?

Reviewer #1: Yes

Reviewer #2: Yes

4. Is the manuscript presented in an intelligible fashion and written in standard English?

Reviewer #1: Yes

Reviewer #2: Yes

5. Review Comments to the Author

Reviewer #1: Peer Review Report for Manuscript PONE-D-24-47138

1. Is the manuscript technically sound, and do the data support the conclusions?

The study presents a well-structured investigation into the use of transthoracic echocardiography for early detection of left ventricular dysfunction in a rat model of chronic renal failure with an arteriovenous fistula. The methodology is clearly outlined, and the experiments appear to have been conducted rigorously with appropriate controls. The statistical analysis is generally sound, and the data seem to support the conclusions. However, additional clarification on sample size justification and power analysis would strengthen the reliability of the findings.

One area of concern is the limited discussion of potential confounding factors, such as sex-based differences and possible variability in cardiac function assessments. Additionally, the inclusion of a more detailed comparison with existing literature could provide stronger contextual support for the study’s claims.

2. Has the statistical analysis been performed appropriately and rigorously?

The manuscript employs standard statistical methods, including ANOVA and t-tests, to compare groups. However, while p-values are provided, effect sizes and confidence intervals should be reported to offer a clearer interpretation of the data’s clinical relevance. A justification for the sample size should be explicitly stated to confirm that the study is sufficiently powered to detect meaningful differences.

3. Have the authors made all data underlying the findings in their manuscript fully available?

The manuscript states that all relevant data are available within the article. However, to comply with PLOS ONE’s data-sharing policies, the authors should consider depositing raw data in a public repository and providing a DOI or accession number for transparency. Additionally, figures and tables should include access to numerical datasets where applicable.

4. Is the manuscript presented in an intelligible fashion and written in standard English?

The manuscript is well-written and generally clear. However, there are minor grammatical errors and awkward phrasing that should be addressed in revision. Some sections, particularly the results and discussion, would benefit from improved flow and organization to enhance readability. A thorough language edit is recommended to ensure clarity and precision.

Additional Comments:

Ethical considerations: The study adheres to animal research ethics guidelines, but the methods section should explicitly state whether humane endpoints were used to minimize animal suffering.

Publication ethics: No concerns about dual publication or research misconduct were identified.

Figures and tables: Some figure legends require more detailed explanations to ensure the data are easily interpretable.

Recommendation:

The study presents valuable insights into early cardiac dysfunction detection using echocardiography in a rat model of chronic renal failure. With improvements in statistical reporting, language clarity, and data transparency, the manuscript would be suitable for publication. Minor revisions are recommended.

Additional comments:

The manuscript should strictly follow the PLOS ONE submission guidelines, including structure, formatting, and citation style.

Experimental Design and Clarity:

The rationale for selecting female rats exclusively should be explained or justified, given potential sex-specific differences in cardiac remodeling.

The choice of a 6-week endpoint for the study is understandable due to mortality concerns, but it limits the ability to assess long-term changes. This limitation should be clearly emphasized in the discussion.

Data Presentation:

Figures and tables are critical for supporting key findings. Ensure all figures (e.g., histopathological images, strain imaging graphs) are high quality and provide enough detail to validate the claims. Some legends are unclear and should be revised for better comprehension.

Reviewer #2: This study presents a novel approach to assess left ventricular dysfunction (LVD) in rats with chronic renal failure (CRI) and arteriovenous fistula (AVF) using advanced imaging techniques. The researchers identified early functional changes by means of global longitudinal strain (GLS), global circumferential strain (GCS) and E/DT ratio, parameters that proved to be more sensitive than conventional echocardiography parameters. The results show continuous myocardial damage over time, underlining the potential of these innovative methods in detecting early cardiac dysfunction in a clinically relevant condition.

The study is well written and structured with adequate statistical analysis. Despite some limitations, the methodology is sound, and the conclusions are well justified, making this research a valuable contribution to the field.

Major comments:

- In the results better explain the results of EF and Strain, because we have an increase in EF and a decrease in Strain between the groups.

Minor Comments:

- Lines 123-123: 4 chamber view or 5 chamber view to obtain the LAD measurements.

- Lines 130-131: How do you obtain the RVCO? You said you don’t evaluate the tight heart for the artifact.

- In the tables will be useful to put the number of animals in each group.

- I suggest to put the values of the Systolic variables in the tables (LVESD, LVESV), the EF ad FS depends from them.

- Figure 2-3: the legend does not specify which group the bullette is from.

- For histopathological images, you have to put the same area for comparison. You have a section with the fibre in longitudinal axis and the comparison in transverse axis.

I suggest you improve the quality of the images.

6. PLOS authors have the option to publish the peer review history of their article (what does this mean? ). If published, this will include your full peer review and any attached files.

**Do you want your identity to be public for this peer review?** For information about this choice, including consent withdrawal, please see our Privacy Policy .

Reviewer #1: No

Reviewer #2: No

---

## [Author Response · Author response to Decision Letter 1]

24 Apr 2025

Dear Editor,

We really appreciate the critical reading of our manuscript entitled “Application of Transthoracic Echocardiography to Assess the Dynamic Evolution of Early Cardiac Damage Induced by Abdominal Aorta-Inferior Vena Cava Fistula in Rats with Chronic Renal Failure”, and we thank you for all the valuable suggestions received from the editors and reviewers. We have carefully considered the comments and have revised the manuscript accordingly. The responses to the comments are listed one by one as follows (please see below)�Also, a professional English proofreading company has carefully revised the manuscript..

We are looking forward to hearing about your final decision.

Sincerely,

Haiyan Wang

Response to editor

1. When submitting your revision, we need you to address these additional requirements. Please ensure that your manuscript meets PLOS ONE's style requirements, including those for file naming. The PLOS ONE style templates can be found at https://journals.plos.org/plosone/s/file?id=wjVg/PLOSOne_formatting_sample_main_body.pdf and https://journals.plos.org/plosone/s/file?id=ba62/PLOSOne_formatting_sample_title_authors_affiliations.pdf

Replay: Thank you for your kind suggestion. We have carefully reviewed our manuscript and revised according to the PLOS ONE's style requirements.

2. To comply with PLOS ONE submissions requirements, in your Methods section, please provide additional information regarding the experiments involving animals and ensure you have included details on (1) methods of sacrifice, and (2) efforts to alleviate suffering.

Replay: Thank you for your comment and kind suggestion. We have provided more details regarding humane euthanasia of animals and measures to minimize animal suffering at page 4 line 69-83.

3. Thank you for stating the following financial disclosure: [This work was supported by: Natural Science Foundation of Shandong Province (ZR2023MH041)].

Reply: Thank you for the kind suggestion. We have added the role of funders at page 30-31, line 457- 461.

Additional Editor Comments:

Major issues should be carefully addressed by the authors:

1) Title: add "...in female rats..." and discuss in the main text the choice of female rats.

Reply : Thank you for your comment. We have added the female rats in title, and discussed the choice of female rats at page 29 line 425-432.

2) "All surgery was performed under isoflurane...": please add concentration of the dose.

Reply : We have added the concentration of the isoflurane at page 4 line 75.

3) Add information on assay used to measure each parameter.

Reply :The information on assay used to measure each parameter have been added at page 6 line 114-124.

4) Add reference supporting the CRF model and add more methodological details on the model.

Reply : The reference and more methodological details about establishing the CRF model were added at section of Establishment of the Chronic Renal Failure Model (page 4-5 , line 86-92)[1].

5) Is echocardiography performed in awaken or anesthetized animals?

Reply : All echocardiography was performed on rats under anesthetized.

We have specified at page 6, line 126.

6) TOMTEC image analysis software: add details on the firm.

Reply :TOMTEC image analysis software (version 2.31; TOMTEC Imaging Systems GmbH, Germany)

7) Table 1: Is the CRF model without significant changes in BUN and serum creatinine a clinically relevant model? Please discuss on it mentioning proper reference.

Reply :The present study aimed to evaluate the influence of AVF on myocardium based on the CRF model. And all the values of BUN and Scr presented in the manuscript was detected after the successful CRF model was verified, while the feeding 0.75% adenine was stopped. And the renal function at the three groups didn’t present significant changes, in accordance with previous reports. [2]

8) Table 3: add data on heart rate and cardiac mass.

Reply : Thank you for kind advice, the data on heart rate and left ventricular mass have been added at Table 3.

9) Echocardiographic parameters should be normalized on body weight

Reply : To minimize the confounding effect, the weight of rats enrolled in study was controlled in a narrow range (250–300g). No significant difference among groups in body weight (P > 0.05), while eliminating the body weight-based difference on results of echocardiographic parameters. And the details of body weight have been added at Table 1.

10) Figure legends: improve the text by adding sample size.

Reply : Thank you for your suggestion�we have added sample at each Figure legends.

11) Since CRF leads to hypertrophic cardiomyocytes, the authors should measure size of cardiomyocytes. Finally, data should be discussed in the light of previous unmentioned study(10.1126/sciadv.adj4846).

Reply : Thank you for your professional suggestion. It’s absolutely great meaning for our study if the size of cardiomyocytes can be measured as you recommended. However, due to limitation in technology and research funding, measurements of cardiomyocyte dimensions were not performed. In this study, echocardiographic parameter including LVEDD, LVEDV, IVS, LVPW, and LVM can provided indirect evidence of significant left ventricular remodeling in the AVF group. In future studies, measuring cardiomyocyte size using the method you mentioned should be adopted to address this limitation in the presented study. We have explained this question at the Limitations section (Page 30, Lines 441-443) and incorporated this reference .

Reviewer's Responses to Questions

Comments to the Author

1. Is the manuscript technically sound, and do the data support the conclusions?

Reviewer #1: Yes

Reviewer #2: Yes

2. Has the statistical analysis been performed appropriately and rigorously?

Reviewer #1: Yes

Reviewer #2: Yes

3. Have the authors made all data underlying the findings in their manuscript fully available?

Reviewer #1: Yes

Reviewer #2: Yes

4. Is the manuscript presented in an intelligible fashion and written in standard English?

Reviewer #1: Yes

Reviewer #2: Yes

5. Review Comments to the Author

Reviewer #1: Peer Review Report for Manuscript PONE-D-24-47138

1. Is the manuscript technically sound, and do the data support the conclusions?

The study presents a well-structured investigation into the use of transthoracic echocardiography for early detection of left ventricular dysfunction in a rat model of chronic renal failure with an arteriovenous fistula. The methodology is clearly outlined, and the experiments appear to have been conducted rigorously with appropriate controls. The statistical analysis is generally sound, and the data seem to support the conclusions. However, additional clarification on sample size justification and power analysis would strengthen the reliability of the findings.

One area of concern is the limited discussion of potential confounding factors, such as sex-based differences and possible variability in cardiac function assessments. Additionally, the inclusion of a more detailed comparison with existing literature could provide stronger contextual support for the study’s claims.

Reply: Thank you for your kind advices, all of us greatly appreciated your patient evaluation for our article and your advices were meaningful and enlightening for our study.

We have carefully addressed the concerns as follows:

Firstly, the sex-based differences and variability in cardiac function was well established and known according to numerous previous studies. And using female rat may generate potential confounding factors for our results[3]. An using female rats in this study due to the following considerations: present study based on the our prior work which focusing on the sex-specific responses in building AVF model[1]. As a part of research in our team, where female rats were used to maintain methodological continuity. Certainly, we fully acknowledged the importance of sex-based differences in cardiac pathophysiology. Future studies will explicitly include male rats to address this critical aspect.

We have improved the discuss about the sex-related confounding factor, and further explain the using female rat in our study at page 29 line 425-432.

Secondly, the measure to minimize the possible variability in cardiac function assessments has been specialized at the Methods sections (page 8 line 162-165).

Finally, we have added the existing literatures and compared these stydies with our research at page 25 line 345-350.

2. Has the statistical analysis been performed appropriately and rigorously?

The manuscript employs standard statistical methods, including ANOVA and t-tests, to compare groups. However, while p-values are provided, effect sizes and confidence intervals should be reported to offer a clearer interpretation of the data’s clinical relevance. A justification for the sample size should be explicitly stated to confirm that the study is sufficiently powered to detect meaningful differences.

Reply: Thank you for comments. We have added the values of effect sizes and confidence intervals in the part of results at Table 1-3. And the justification for sample size was also stated.

"In addition to null hypothesis testing, all between-group comparisons now report effect sizes with 95% confidence intervals. Cohen's d was calculated for t-tests (d = 0.2: small, 0.5: medium, 0.8: large) [1], while partial eta-squared (η²) was used for ANOVA (η² > 0.01: small, >0.06: medium, >0.14: large.

According to the success rates of establishing both the CRF and AVF rat models in the previous study[1], along with the requirement to sacrifice 3 rats at each time point. Meanwhile, minimizing the number of animals sacrificed should not be neglected to for the reason of animal ethics principles. Finally, 45 rats (15 per group) were enrolled to ensure statistical power.

3. Have the authors made all data underlying the findings in their manuscript fully available?

The manuscript states that all relevant data are available within the article. However, to comply with PLOS ONE’s data-sharing policies, the authors should consider depositing raw data in a public repository and providing a DOI or accession number for transparency. Additionally, figures and tables should include access to numerical datasets where applicable.

Reply All relevant data are within the manuscript and its Supporting Information files.

4. Is the manuscript presented in an intelligible fashion and written in standard English?

The manuscript is well-written and generally clear. However, there are minor grammatical errors and awkward phrasing that should be addressed in revision. Some sections, particularly the results and discussion, would benefit from improved flow and organization to enhance readability. A thorough language edit is recommended to ensure clarity and precision.

Reply: Thank you for your careful reading and kind suggestion. The revised manuscript has been revised by a professional English-language editor, and the grammatical errors and awkward phrasing have been addressed. We have improved the flow and organization of manuscripts to ensure the readability, particularly section of results and discussion.

Additional Comments:

Ethical considerations: The study adheres to animal research ethics guidelines, but the methods section should explicitly state whether humane endpoints were used to minimize animal suffering.

Reply Thank you for comment. The methods of humane euthanasia was added at page 4, line 81-83.

Publication ethics: No concerns about dual publication or research misconduct were identified.

Reply: We have added the statement of publication ethics at page 31 line 467.

Figures and tables: Some figure legends require more detailed explanations to ensure the data are easily interpretable.

Reply: Thank you for kind advice, we have revised and improved the figure legends with more detailed information to ensure the data are easily interpretable.

Recommendation:

The study presents valuable insights into early cardiac dysfunction detection using echocardiography in a rat model of chronic renal failure. With improvements in statistical reporting, language clarity, and data transparency, the manuscript would be suitable for publication. Minor revisions are recommended.

Additional comments:

The manuscript should strictly follow the PLOS ONE submission guidelines, including structure, formatting, and citation style.

Reply:

We have carefully checked and revised the manuscript to ensure full compliance with the PLOS ONE submission guidelines.

Experimental Design and Clarity:

The rationale for selecting female rats exclusively should be explained or justified, given potential sex-specific differences in cardiac remodeling.

The choice of a 6-week endpoint for the study is understandable due to mortality concerns, but it limits the ability to assess long-term changes. This limitation should be clearly emphasized in the discussion.

Reply: We sincerely appreciated the reviewer’s careful reading and professional comments. Your suggestion provides large help in further improving the manuscript and our future research. As you mentioned, it’s well established that the course of cardiac remodeling existed potential sex-specific differences [3], which may generate potential confounding factors on results. An using female rats in this study was due to the following considerations: present study based on the prior work of our team which focusing on the sex-specific responses in building AVF mode[1]. As part of our research, where female rats were used to maintain methodological continuity. However, we fully acknowledged the importance of sex-based differences in cardiac pathophysiology. Future studies will explicitly include male rats to address this critical aspect.

Considering to the animal welfare of reducing animal suffering, and reducing mortality in the following, the study endpoint was set at 6 weeks, but it limits the ability to assess long-term changes in cardiac damage due to AVF.

Data Presentation:

Figures and tables are critical for supporting key findings. Ensure all figures (e.g., histopathological images, strain imaging graphs) are high quality and provide enough detail to valid

---

## [Decision Letter · Decision Letter 1]

4 May 2025

Application of Transthoracic Echocardiography to Assess the Dynamic Evolution of Early Cardiac Damage Induced by Abdominal Aorta-Inferior Vena Cava Fistula in Rats with Chronic Renal Failure

PONE-D-24-47138R1

Dear Dr. Wang,

We’re pleased to inform you that your manuscript has been judged scientifically suitable for publication and will be formally accepted for publication once it meets all outstanding technical requirements.

Kind regards,

Vincenzo Lionetti, M.D., PhD

Academic Editor

PLOS ONE

Additional Editor Comments (optional):

Reviewers' comments:

Reviewer's Responses to Questions

**Comments to the Author**

1. If the authors have adequately addressed your comments raised in a previous round of review and you feel that this manuscript is now acceptable for publication, you may indicate that here to bypass the “Comments to the Author” section, enter your conflict of interest statement in the “Confidential to Editor” section, and submit your "Accept" recommendation.

Reviewer #2: (No Response)

2. Is the manuscript technically sound, and do the data support the conclusions?

Reviewer #2: Yes

3. Has the statistical analysis been performed appropriately and rigorously? 

Reviewer #2: Yes

4. Have the authors made all data underlying the findings in their manuscript fully available?

Reviewer #2: Yes

5. Is the manuscript presented in an intelligible fashion and written in standard English?

Reviewer #2: Yes

6. Review Comments to the Author

Reviewer #2: (No Response)

7. PLOS authors have the option to publish the peer review history of their article (what does this mean? ). If published, this will include your full peer review and any attached files.

**Do you want your identity to be public for this peer review?** For information about this choice, including consent withdrawal, please see our Privacy Policy .

Reviewer #2: No

---

## [Editor Report · Acceptance letter]

PONE-D-24-47138R1

PLOS ONE

Dear Dr. Wang,

I'm pleased to inform you that your manuscript has been deemed suitable for publication in PLOS ONE. Congratulations! Your manuscript is now being handed over to our production team.

Kind regards,

on behalf of

Prof. Vincenzo Lionetti

Academic Editor

PLOS ONE